# Thyroid carcinoma in children, adolescents, and young adults in Brazil: A report from 11 population-based cancer registries

**Rejane de Souza Reis[1], Gemma Gatta[2]ʘ, Beatriz de Camargo[3]ʘ ***

**1** Hospital Fundação do Câncer, Rio de Janeiro, RJ, Brazil, **2** Fondazione IRCCS Istituto Nazionale Tumori, Milan, Italy, **3** Research Center, Instituto Nacional de Cancer, Rio de Janeiro, RJ, Brazil

ʘ These authors contributed equally to this work.
* bdecamar@terra.com.br

## Abstract

### Background

The increasing incidence of thyroid cancer has been described worldwide. Overdiagnosis, improved imaging, and increased environmental risk factors have contributed to the rising incidence. The objective of this study was to analyze the population incidence rate and trends during the period of 2000–2013 in children, adolescents and young adults (AYAs) in Brazil.

### Methods

Data were extracted from 11 population-based cancer registries (PBCRs) encompassing the five geographic regions of Brazil. Incidence rates per million in children (0–14) and AYAs (15–39) according to world population were analyzed according to sex, age, and type of carcinoma. Incidence trends were evaluated using joinpoint regression.

### Results

During 2000 to 2013, we identified 11,081 children and AYAs (0–39 years) with thyroid carcinoma in 11 PBCRs, with an age-adjusted incidence rate (AAIR) of 42 cases per million. Females had a higher AAIR of 66 cases per million versus 14 cases per million in males. Age-specific incidence rate (ASR) increased with age. Geographic variation was also observed; the Midwest and Southeast regions had the highest ASR in all age groups. The lowest ASR in all age groups was seen in the North region. Papillary subtype was the most common. Overall, the incidence rates in children and AYAs significantly increased from 0.2 in 2000 to 2.8 in 2013 and from 47.1 to 115.3, respectively, with an annual average percent change of 18.8 [8.1; 30.6] for children and 7.9 [CI 5.6; 10.3] for AYAs.

### Conclusions

Rates of thyroid cancer, particularly the papillary subtype, are steadily increasing in children and AYAs, especially among females. There are variations among geographic areas. This

**Data Availability Statement:** All relevant data are within the manuscript and its Supporting Information files.

**Funding:** The authors received no specific funding for this work.

**Competing interests:** The authors have declare no competing interests.

increased incidence is unlikely to be explained by screening, as children less than 14 years of age do not typically undergo medical surveillance. Environmental risk factors must be investigated.

## Introduction

Thyroid cancer in children and adolescents is one of the most common rare tumors, as defined by TREP in Brazil [1]. According to a recent cancer incidence in five continents (CI5C) monograph, Brazil has the highest incidence rates within South and Central America [2]. An increased incidence of thyroid cancer in adults and childhood has been reported in several countries [3, 4, 5]. However, the magnitude of the effects of overdiagnosis and increased environmental risk factors on this higher incidence is not well known. In children, thyroid nodule screening is less likely to occur and recent data from North American Association of Central Cancer Registries showed that large and late-stage thyroid carcinomas are increasing over time [6].

A recent analysis of rare tumors in children and adolescents (0–19 years old) in Brazil showed different incidence rates of thyroid carcinomas in Brazilian geographic regions varying from 1.87 per million in the North region to 6.48 per million in the Southeast region. Incidence rates increase with age and were found to be the highest among adolescents (15–19 years old) [1], Incidence rates in Latin American in adolescents and young adults (AYAs) are the most common cancer in females was thyroid cancer, with rates ranging from 0.9 to 10.0/ 100,000 [7]. It is well described that the increase is attributable to an increase in incidence of the histologic subtype papillary thyroid cancer. Clinicopathological characteristics of young patients with thyroid carcinomas seem to differ from those of adults. The lymph node metastasis rate is significantly higher, but distant metastases are lower in AYAs [8]. Sociodemographic disparities are described as prognostic factors affecting survival among the AYA group, with a worse prognosis in socioeconomically deprived groups [9].

Thyroid cancer is a common second malignancy in children with cancer [10]. The cause of the increasing incidence remains to be established, but enhanced screening diagnostic and others factors should be investigated. Few risk factors have been described. Exposure to ionizing radiation is the most established risk factor. Irradiation in childhood or young adulthood increases the risk of nodules and papillary thyroid cancer [11]. The prevalence of incidental thyroid abnormalities detected by ultrasound in **Korean** adults is about 13–67% [12]. In children, few reports have studied the prevalence of ultrasound-detected findings in the thyroid, with the exception of screening programs conducted around Chernobyl [13]. A survey including thyroid ultrasound examinations for children aged ≤18 years was conducted in Japanese children unaffected by radioactive material from the Fukushima Nuclear Power Plant accident, and thyroid nodules were identified in 1.6% of participants; 99% were classified as 'no further examination required' [14, 15]. Neck ultrasound done for other reasons detected incidental thyroid abnormalities in 18% of children, and the majority were cystic lesions. No malignancy was found in this group of patients [16]. In addition, increased iodine intake, obesity, and higher birth weight have also been described as potential risk factors [17, 18, 19].

The objective of this study was to analyze trends of incidence rates in thyroid carcinomas among children and AYAs (0–39 years) in Brazil, and provide suggestions for further investigations into descriptive and etiological epidemiology.

## Materials and methods

Incidence data were obtained from 11 population-based cancer registries (PBCRs) encompassing the five geographic regions of Brazil: North (Belém and Palmas), Northeast (Aracaju, João Pessoa and Recife), Midwest (Goiânia), Southeast (Barretos, Belo Horizonte, Jahu and São Paulo), and South (Curitiba). Table 1 shows the population coverage of children (0–14 years old) and AYAs (15–39 years old) for each city, providing PBCR data in five geographic regions during the period of coverage. (Table 1)

The International Classification of Disease for Oncology, second and third editions [20, 21] was used according to histological group papillary, follicular, other carcinomas, and not otherwise specified (NOS) carcinomas. Other carcinomas included all other carcinoma subtypes found at the 11 PBCRs (Table 2).

Table 3 presents data quality indicators. NOS morphology codes (8000–8010) were higher than 10% in two PBCRs. Only four PBCRs had different than zero, as ascertained by death certificate only. The rate of microscopically verified cases was over 85% in 10 PBCRs, and only 1 had a rate below 75%.

To estimate crude incidence rates, we divided the numbers of incident cases during the period of diagnosis by the corresponding person-years lived within the general population during the same period for each PBCR. Age-adjusted incidence rates (AAIRs) were estimated by the direct method, using the world population estimated for gender and age groups, and were analyzed according to age group [22]. Age-specific incidence rates (ASRs) per million were analyzed according to age at diagnosis, with the age range stratified into seven groups: 0–9, 10–14, 15–19, 20–24, 25–29, 30–34, and 35–39 years of age. Incidence trends were estimated with annual average percent change (AAPC) and corresponding confidence intervals (CIs) with Joinpoint software (version 4.6.0.0). P values less than 0.05 were considered statistically significant [23].

All data used for this analysis were public access data and ethical approval was not required.

**Table 1. Population of children and AYAs in five Brazilian geographic regions of the 11 PBCR, 2000–2013.**

| Local | Age-group | |
|---|---|---|
| | **0–14** | **15–39** |
| **North region** | | |
| Belém | 7,032,422 | 12,088,341 |
| Palmas | 839,998 | 1,432,022 |
| **Northeast region** | | |
| Aracaju | 1,855,000 | 3,388,063 |
| João Pessoa | 2,375,388 | 4,285,109 |
| Recife | 5,042,777 | 9,268,313 |
| **Midwest region** | | |
| Goiânia | 4,031,550 | 8,023,995 |
| **Southeast region** | | |
| Belo Horizonte | 7,325,583 | 14,613,982 |
| Barretos | 1,335,602 | 2,354,375 |
| Jahu | 387,548 | 733,590 |
| São Paulo | 35,610,705 | 66,924,283 |
| **South region** | | |
| Curitiba | 5,535,231 | 10,821,232 |

Source: MP/Fundação Instituto Brasileiro de Geografia e Estatística–IBGE

**Table 2. Thyroid carcinomas studied, with ICD-O2 and ICD-O3 topography and morphology codes.**

| Classification | Morphology codes | Topography code |
|---|---|---|
| Papillary | 8050, 8052, 8130, 8260, 8340–8344, 8450, 8452 | C73 |
| Follicular | 8290, 8330–8332, 8335 | |
| Others carcinomas* | 8020, 8021, 8030, 8070, 8074, 8140, 8190, 8200, 8230, 8246, 8262, 8310, 8337, 8345, 8350, 8380, 8430, 8510, 8511, 8550 | |
| Carcinomas, NOS | 8000–8010 | |

* Others carcinomas includes all others carcinomas subtypes found in 11 PBCR

Abbreviation: ICD-02; ICD-03: International Classification of Disease for Oncology, 2$^{nd}$/ 3$^{rd}$ Edition; NOS: not other specified

## Results

Between 2000 and 2013, we identified a Brazilian pool from 11 PBCRs of 11.081 children and AYAs (0–39 years) with thyroid carcinoma (AAIR of 41.64 cases per million). Table 4 presents the number of cases and ASRs per million incidence rates according to gender by geographic region. ASR increased with age, and females had a higher ASR in all age groups. Geographic variation was observed and the Midwest and Southeast regions had the highest ASRs in all age groups, although among children 10–14 years of age, the South region had a higher ASR. The lowest ASR in all age groups was found in the North region. ASR in all age groups and PBCR can be seen in Supporting additional table (S1 Table). There are an increase with age and female has a higher ASR in all age groups.

Fig 1 shows the ratio distribution by histologic subtypes according to age group. The most common histologic subtype in all 11 PBCRs was papillary, accounting for more than 70%. The

**Table 3. Data quality indicators from 11 Brazilian PBCR in cases aged <39 years diagnosed during the period 2000–2013.**

| PBCR | Total number of cases | NOS% | DCO% | MV% |
|---|---|---|---|---|
| North | | | | |
| Belém | 156 | 8.3 | 0.6 | 97.4 |
| Palmas | 22 | 0.0 | 0.0 | 90.9 |
| Northeast | | | | |
| Aracaju | 483 | 0.6 | 0.0 | 99.8 |
| João Pessoa | 120 | 12.5 | 0.0 | 97.5 |
| Recife | 220 | 7.3 | 0.5 | 96.4 |
| Midwest | | | | |
| Goiânia | 591 | 1.2 | 0.0 | 73.4 |
| Southeast | | | | |
| Barretos | 111 | 7.2 | 0.0 | 100,0 |
| Belo Horizonte | 377 | 0.0 | 0.3 | 99.5 |
| Jahu | 36 | 0.0 | 0.0 | 100,0 |
| São Paulo | 8513 | 14.5 | 0.0 | 88.7 |
| **South** | | | | |
| Curitiba | 452 | 1.5 | 0.2 | 98.0 |

Abbreviations: PBCR: population-based cancer registry; NOS: Not otherwise specified; DCO: Death certificate only; MV: Microscopically verification

**Table 4. Numbers of cases and incidence rates of thyroid carcinomas in five Brazilian regions by gender in cases aged <40 years and diagnosed during the period 2000–2013.**

| Gender | Local | Age-group | | | | | | | | |
|---|---|---|---|---|---|---|---|---|---|---|
| | | 0–14 | | | 15–39 | | | <40 | | |
| | | N | ASR | AAIR* | N | ASR | AAIR* | N | ASR | AAIR* |
| Total | North | 6 | 0.76 | 0.67 | 172 | 12.72 | 12.55 | 178 | 8.32 | 7.13 |
| | Northeast | 10 | 1.08 | 0.94 | 813 | 47.99 | 46.00 | 823 | 31.39 | 25.46 |
| | Midwest | ** | 1.24 | 1.02 | 586 | 73.03 | 70.14 | 591 | 49.02 | 38.63 |
| | Southeast | 103 | 2.24 | 1.99 | 8 934 | 102.65 | 94.92 | 9 037 | 67.96 | 52.56 |
| | South | 12 | 2.17 | 1.77 | 440 | 40.66 | 38.03 | 452 | 27.63 | 21.50 |
| | Brazilian pool | 136 | 1.87 | 1.64 | 10 945 | 80.28 | 75.14 | 11 081 | 53.02 | 41.64 |
| Females | North | ** | 1.28 | 1.10 | 94 | 13.25 | 20.21 | 108 | 9.82 | 11.50 |
| | Northeast | 7 | 1.53 | 1.36 | 693 | 77.61 | 73.21 | 700 | 51.81 | 40.45 |
| | Midwest | ** | 1.51 | 1.23 | 487 | 116.46 | 110.98 | 490 | 79.36 | 60.95 |
| | Southeast | 72 | 3.08 | 2.77 | 7 507 | 163.46 | 150.20 | 7 579 | 109.39 | 82.98 |
| | South | 8 | 2.94 | 2.93 | 377 | 68.11 | 63.12 | 385 | 46.63 | 35.43 |
| | Brazilian pool | 95 | 2.60 | 2.29 | 9 210 | 128.51 | 119.36 | 9 305 | 85.98 | 65.99 |
| Males | North | ** | 0.25 | 0.24 | 26 | 4.05 | 4.04 | 27 | 2.60 | 2.31 |
| | Northeast | ** | 0.64 | 0.52 | 120 | 14.98 | 14.71 | 123 | 9.68 | 8.24 |
| | Midwest | ** | 0.98 | 0.81 | 99 | 25.77 | 25.15 | 101 | 17.17 | 14.05 |
| | Southeast | 31 | 1.37 | 1.20 | 1 427 | 34.71 | 32.38 | 1 458 | 22.89 | 18.16 |
| | South | ** | 1.42 | 1.17 | 63 | 11.92 | 11.34 | 67 | 8.27 | 6.70 |
| | Brazilian pool | 41 | 1.14 | 0.98 | 1 735 | 26.83 | 25.41 | 1 776 | 17.62 | 14.27 |

* less than 5 cases

Abbreviations: N: number of cases; ASR: age-specific rate; AAIR: Age-adjusted incidence rate; PBCR population-based cancer registries

non-specified group was the least common among adolescents (15–19 years old), and the group classified as other carcinomas was most common in the 10 to 14-year-old age group.

In Table 5, morphologic subtypes according to gender shows that females and AYAs (15–39 years old) and papillary subtype had the highest CRs and AAIRs.

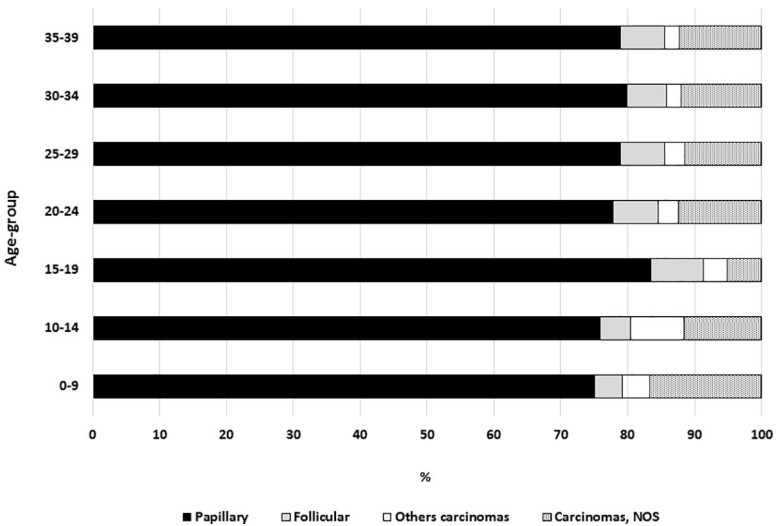

**Fig 1. Percentage distribution by histologic subtypes according to age group.**

**Table 5. Numbers of cases and crude and age-adjusted incidence rate of thyroid carcinoma according to age group, gender and histologic subtypes.**

| Age-group | Histologic type | Gender | | | | | | | | | F:M ratio |
|---|---|---|---|---|---|---|---|---|---|---|---|
| | | Female | | | Male | | | Total | | | |
| | | N | CR | AAIR* | N | CR | AAIR* | N | CR | AAIR* | |
| **0–14** | Papillary | 77 | 2.11 | 1.86 | 26 | 0.72 | 0.62 | 103 | 1.4 | 1.3 | 2.93 |
| | Follicular | ** | 0.08 | 0.07 | ** | 0.08 | 0.07 | 6 | 0.1 | 0.1 | 1.00 |
| | Others carcinomas | 6 | 0.16 | 0.15 | ** | 0.11 | 0.09 | 10 | 0.1 | 0.1 | 1.45 |
| | Carcinomas, NOS | 9 | 0.25 | 0.21 | 8 | 0.22 | 0.20 | 17 | 0.2 | 0.2 | 1.14 |
| | **Total** | **95** | **2.60** | **2.29** | **41** | **1.14** | **0.98** | **136** | **1.87** | **1.64** | **2.28** |
| **15–39** | Papillary | 7 292 | 101.75 | 94.54 | 1 389 | 21.48 | 20.33 | 8 681 | 63.67 | 59.61 | 4.74 |
| | Follicular | 620 | 8.65 | 8.10 | 82 | 1.27 | 1.21 | 702 | 5.15 | 4.85 | 6.81 |
| | Others carcinomas | 206 | 2.87 | 2.68 | 71 | 1.10 | 1.08 | 277 | 2.03 | 1.93 | 2.61 |
| | Carcinomas, NOS | 1 092 | 15.24 | 14.04 | 193 | 2.98 | 2.80 | 1285 | 9.42 | 8.75 | 5.11 |
| | **Total** | **9 210** | **128.51** | **119.36** | **1 735** | **26.83** | **25.41** | **10 945** | **80.28** | **75.14** | **4.79** |
| **0–39** | Papillary | 7 369 | 68.09 | 52.29 | 1 415 | 14.04 | 11.34 | 8 784 | 42.03 | 33.00 | 4.85 |
| | Follicular | 623 | 5.76 | 4.44 | 85 | 0.84 | 0.69 | 708 | 3.39 | 2.67 | 6.86 |
| | Others carcinomas | 212 | 1.96 | 1.52 | 75 | 0.74 | 0.63 | 287 | 1.37 | 1.10 | 2.65 |
| | Carcinomas, NOS | 1 101 | 10.17 | 7.74 | 201 | 1.99 | 1.61 | 1 302 | 6.23 | 4.86 | 5.11 |
| | **Total** | **9 305** | **85.98** | **65.99** | **1 776** | **17.62** | **14.27** | **11 081** | **53.02** | **41.64** | **4.88** |

* less than 5 cases

Abbreviations: N: number of cases; CR: crude rate; AAIR: age-adjusted incidence rate

The female:male ratio was 2.28 among children 0–14 years old and 4.79 for AYAs (15–39 years old). Rates increased for both age groups, children and AYAs. Among children, less than 15 years old, an increase in papillary histologic subtype was only seen in the São Paulo PBCR. Among AYAs (15–39 years), there was a significant increase in different Brazilian geographic regions. Belo Horizonte, which is located in the Midwest region, had the highest rate of increase (AAPC 13.8 [7.4; 20.6] (Table 6).

Overall, the incidence rates significantly increased in children (0–14) and AYAs (15–39), from 0.19 in 2000 to 2.80 in 2013 and from 47.11 in 2000 to 115.30 in 2013, respectively (Fig 2). In the two age groups, the incidence of papillary subtype increased from 0.35 in 2001 to 2.19 in 2013 for children and from 31.48 in 2000 to 100.39 in 2013 for AYAs. For the follicular subtype, the incidence was stable during the study period for AYAs only (Fig 2).

## Discussion

To the best of our knowledge, this is the first descriptive analysis of Brazilian PBCRs regarding thyroid carcinomas in children and AYAs. During the period of 2000–2013, the rates of thyroid carcinoma significantly increased among children and AYAs in Brazil. The incidence rate for children 0–9 years old was 0.51 per million, increasing with age and predominating in females. Children 10–14 years old had about a 9-fold greater incidence than younger children 0–9 years old. AYAs had a 4-fold greater incidence than children (0–14 years). There was a female preponderance at all ages. Incidence rates were 4.4, 17.5, 39.2, and 76.7 per million for the 10–14, 15–19, 20–24, and 25–29 year age groups, respectively, similar to the United States Surveillance, Epidemiology, and End Results (US SEER) data [24] during 1984–2010.

The increasing incidence rate (1973–2002) has been reported worldwide for the total population and does not appear to be geographically dependent [25, 26]. The reasons why the incidence of thyroid carcinoma has been increasing are still a subject of debate. Improved

**Table 6. Trend of thyroid carcinomas incidence by gender, age group and Brazilian region.** Age-adjusted rate and AAPC, period 2000–2013.

| Variable | 0–14 | | | 15–39 | | |
|---|---|---|---|---|---|---|
| | N | AAIR | AAPC (95%CI) | N | AAIR | AAPC (95%CI) |
| Total | 136 | 1.64 | 18.8* (8.1; 30.6) | 10 945 | 75.14 | 7.9* (5.6; 10.3) |
| Gender | | | | | | |
| Female | 95 | 2.29 | 23.2* (4.4; 45.3) | 9 210 | 119.36 | 7.8* (5.5; 10.2) |
| Male | 41 | 0.98 | - | 1 735 | 25.41 | 8.6* (5.2; 12.1) |
| Histologic group | | | | | | |
| Papillary | 103 | 1.25 | 17.6* (8.3; 27.7) | 8 681 | 63.67 | 10.8* (8.6; 13.1) |
| Follicular | 6 | 0.07 | - | 702 | 5.15 | -0.6 (-8.0; 7.4) |
| Brazillian PBCR | | | | | | |
| North Region | | | | | | |
| Belém | ** | 0.63 | - | 151 | 12.28 | 4.2* (0.4; 8.1) |
| Palmas | ** | 1.06 | - | 21 | 14.82 | - |
| Northeast Region | | | | | | |
| Aracaju | ** | 2.20 | - | 478 | 136.39 | 8.6* (4.5; 12.9) |
| João Pessoa | ** | 1.02 | - | 117 | 26.15 | - |
| Recife | ** | 0.44 | - | 218 | 22.87 | 4.3 (-3.1; 12.4) |
| Midwest Region | | | | | | |
| Goiânia | ** | 1.02 | - | 586 | 70.14 | 3.2* (0.5; 5.9) |
| Southeast Region | | | | | | |
| Barretos | ** | 1.33 | - | 109 | 43.86 | 4.5 (-0.7; 9.9) |
| Belo Horizonte | ** | 0.33 | - | 374 | 24.27 | 13.8* (7.4; 20.6) |
| Jahu | 0 | 0.00 | - | 36 | 45.99 | - |
| São Paulo | 98 | 2.37 | 15.9* (8.1; 30.6) | 8 415 | 111.83 | 8.7* (5.7; 11.9) |
| South Region | | | | | | |
| Curitiba | 12 | 1.77 | - | 440 | 38.03 | 1.0 (-1.9; 3.9) |

* less than 5 cases

** $p$-value < 0.05

N: number of cases; AAIR: Age-adjusted incidence rate; AAPC: Annual average percent change; PBCR: population-based cancer registries

diagnosis does not completely explain the incidence trends; rather the increasing incidence is most likely due to a mixture of factors including increased exposure to risk factors. In our series, the overall incidence rates in children (0–14 years old) and AYAs (15–39 years old) significantly increased from 0.2 in 2000 to 2.8 in 2013, and from 47.1 in 2000 to 115.3 in 2013, respectively. The US SEER data did not change in children between 0 and 9 years old, but increasing trends were seen in children from 10 to 14 years old and AYAs, similar to adults [24]. Recently, a significantly increased rate was seen among children and adolescents aged 10 to 19 years [6]. All tumor sizes were noted to increase rates, which cannot be explained by screening only, but may also be affected by environmental and others factors. These data are in accordance with data from adults showing that the observed incidence increased for all tumor sizes, suggesting that early diagnosis is not the only explanation [5]. Unfortunately, we did not have access to stage and size of tumor at diagnosis in this population set.

Our study showed differences among geographic regions. During the period of 1997–2008, data comparing the incidence rates from São Paulo located in the Southeast region of Brazil and from the US SEER showed that Sao Paulo had the highest increase in incidence for all age groups. A previous study suggested that not only a better diagnosis but also differences in iodine nutrition may affect the incidence of this disease [18]. In this report, among children

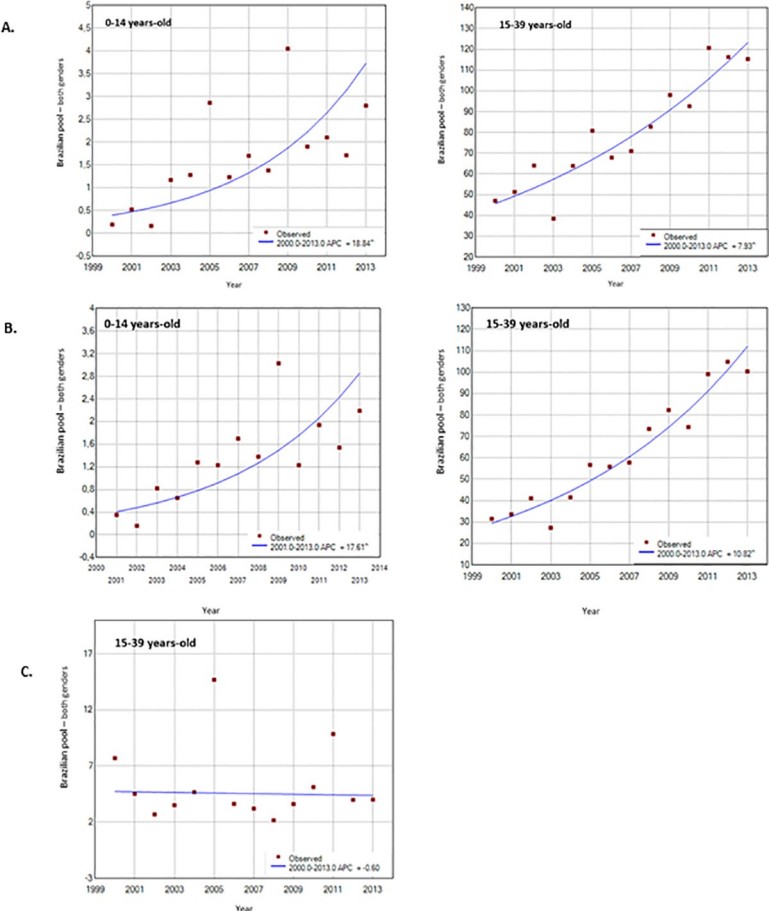

**Fig 2.** Incidence trends of thyroid carcinoma (A) and by histologic subtype, papillary (B) and follicular (C) in children and AYAs.

0–14 years old, we observed a significantly increased incidence rate, possibly attributed to the fact that the Sao Paulo PBCR had the largest number of cases in this age group. Among AYAs of both genders, there was a significantly increased incidence of the papillary subtype. The highest increase in incidence was observed in Aracaju in the Northeast region and in Belo Horizonte and Sao Paulo located in the Southeast region of Brazil.

Few reports has described in children the prevalence of ultrasound-detected findings in the thyroid and routine screening ultrasound in children is not recommended unless they are high risk [13, 14, 15, 27]. Overall, the annual number of imaging diagnostic procedures has rapidly increased in Brazil. Overall ultrasound (any site) increased from 13 million to 309 million between 2000 and 2013. (http://www2.datasus.gov.br/).

A strong association between radiation exposure and risk of thyroid cancer is well documented. After the Chernobyl nuclear accident and more recently after the Fukushima Daiichi Nuclear Power Plant, a significant increase in pediatric thyroid carcinoma has been observed in children living in contaminated areas [13, 15]. In children treated with irradiation for cancer, thyroid carcinoma is one of the most common second malignancies [11]. The use of X-ray and computed tomography (CT) as medical and dental diagnostic sources has increased during the last two decades, and are associated with an increased thyroid cancer risk [28]. Available data has shown increased of dental x-ray of 1% during the period 2008 to 2019 among

children, adolescents and young adults (http://www2.datasus.gov.br/). The use of CT is widespread in pediatric settings and delivers greater radiation doses than conventional X-rays [29]. Dose of radiation and number of CT scans, which can lead to the development of cancer, depends on multiple factors such as the specific type of CT, the patient's age, and gender.

High iodine in diet has been associated with thyroiditis, hypothyroidism, hyperthyroidism, increased risk of papillary carcinoma, and decreased risk of follicular carcinoma [30, 31]. In this study, there was a smaller number of follicular carcinomas and no trends were observed in children and AYAs. Brazil has changed from iodine deficiency to excessive iodine nutritional intake (http://189.28.128.100/dab/docs/portaldab/documentos/pnaisal_relatorio_final.pdf).

The role of iodine in the pathogenesis of thyroid cancer is controversial, and the difference has been documented according to tumor histology. Iodine deficiency is associated with follicular carcinoma, whereas excess iodine increases the risk of papillary type of thyroid cancer [30].

In a recent cohort study from Denmark, birth weight was associated with risk of thyroid cancer and significantly with follicular carcinoma, but these data need to be confirmed [17]. In a Brazilian record linkage analysis regarding birth weight, minor differences were observed among the mean birth weight for tumor types and controls. For the miscellaneous group including the carcinoma group, a risk of 1.37 (CI: 0.71–2.66) was observed for each additional 1000 g of birth weight [32].

Several studies have shown the relationship between overweight and different cancer types including thyroid cancer [19, 33]; however, this has not been shown in children. Childhood obesity has been increasing worldwide. This increasing prevalence has emerged as an important public health problem and has possibly led to the emergence of multiple serious obesity-related comorbidities. The prevalence of obesity among children and adolescents in Brazil has also significantly increased, from 5.4% to 8.5% in girls and boys and from 6.2% to 8.4% in the age group of 5–9 years during 2008–2018. Among adolescents girls, it increased from 3.4% to 7.3% during the same period (http://sisaps.saude.gov.br/sisvan/relatoriopublico/estadonutricional).

Clinical presentation and outcomes in pediatric patients is significantly different compared to adults [34, 35]. Distinct genetic alterations have been suggested. *RET*/PTC rearrangement is the most prevalent genetic alteration found in sporadic and radiation-induced pediatric thyroid papillary carcinomas [36]. In a recent series of 35 Brazilian pediatric patients with thyroid papillary carcinoma, it was confirmed that *RET*/ PTC rearrangement was the most prevalent mutation. The spectrum of mutations was similar to that described in radiation-exposed pediatric papillary carcinoma, suggesting that all thyroid carcinomas may be radiation-induced [36, 37]. Mutation in the *BRAF* gene is the most common abnormality in adult papillary thyroid carcinoma, but is very rare in children [38, 39].

As shown in a previous report, the incidence of thyroid carcinomas was higher for all ages in the Midwest, Southeast, and South regions of Brazil, with the highest percentages observed in white residents [1, 40]. Unfortunately, we do not have reliable data among the PBCRs regarding race/skin color. We observed an increasing incidence of thyroid carcinoma, both papillary and follicular, in children and AYAs in Brazil that can be attributed to several risk factors, such as overdiagnosis; however, others factors must be investigated. Excess iodine intake, obesity, radiation for diagnostic reasons, prevention, and environmental exposure should be investigated.

A major limitation of this study is that there is no information regards size and stage of tumors. Another major limitation is that risk factors as use of radiation, is unknown. We cannot generalize this data to entire Brazil. These 11 PBCR cover approximately 20% of the total Brazilian population.

In conclusion, it was not possible to identify the exact cause or causes of this notable rise in incidence, mainly for papillary carcinomas, in both populations. Increased diagnostic activity may play a role, but is not likely to be the only reason because incidence continues to rise rather than levelling off at some point in time. Future studies especially in Sao Paulo should attempt to retrieve tumor stage and size in order to determine the role of advances in diagnostic accuracy, and to evaluate the impact of iodine prophylaxis on papillary and follicular carcinoma incidence patterns.

## Supporting information

**S1 Table. Numbers of cases and age-specific incidence rates of thyroid carcinomas in five Brazilian regions, 11 PBCR in cases aged <40 years and diagnosed during the period 2000–2013.**
(DOCX)

## Acknowledgments

The authors are grateful to all of the coordinators of the PBCRs in Brazil who contributed the datasets that made this work possible.

## Author Contributions

**Conceptualization:** Rejane de Souza Reis, Gemma Gatta, Beatriz de Camargo.

**Data curation:** Rejane de Souza Reis.

**Formal analysis:** Rejane de Souza Reis, Gemma Gatta, Beatriz de Camargo.

**Methodology:** Rejane de Souza Reis, Gemma Gatta, Beatriz de Camargo.

**Project administration:** Gemma Gatta, Beatriz de Camargo.

**Supervision:** Gemma Gatta, Beatriz de Camargo.

**Writing – original draft:** Rejane de Souza Reis, Gemma Gatta, Beatriz de Camargo.

**Writing – review & editing:** Rejane de Souza Reis, Gemma Gatta, Beatriz de Camargo.

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
