## [Decision Letter · Decision Letter 0]

11 Feb 2020

PONE-D-19-34167

Thyroid carcinoma in children, adolescents, and young adults in Brazil: A report from 11 population-based cancer registries

PLOS ONE

Dear Prof. de Camargo,

Thank you for submitting your manuscript to PLOS ONE. After careful consideration, we feel that it has merit but does not fully meet PLOS ONE’s publication criteria as it currently stands. Therefore, we invite you to submit a revised version of the manuscript that addresses the points raised during the review process.

Please address all comments raised by the reviewers in "Review Comments to the Author section"  fully.

We would appreciate receiving your revised manuscript by Mar 27 2020 11:59PM. To enhance the reproducibility of your results, we recommend that if applicable you deposit your laboratory protocols in protocols.io, where a protocol can be assigned its own identifier (DOI) such that it can be cited independently in the future. For instructions see: http://journals.plos.org/plosone/s/submission-guidelines#loc-laboratory-protocols

We look forward to receiving your revised manuscript.

Kind regards,

Amir Radfar, MD,MPH,MSc,DHSc

Academic Editor

PLOS ONE

Journal Requirements:

2. In the ethics statement in the manuscript and in the online submission form, please provide additional information about the patient records used in your retrospective study, including: a) whether all data were fully anonymized before you accessed them; b) the date range (month and year) during which patients' medical records were accessed; and d) the source of the medical records analyzed in this work (e.g. the names of the 11 population-based cancer registries).

"NO funders had any role in the study design, data collection and analysis, preparation of the manuscript."

4. We note that Figure 1 in your submission contains map images which may be copyrighted. All PLOS content is published under the Creative Commons Attribution License (CC BY 4.0), which means that the manuscript, images, and Supporting Information files will be freely available online, and any third party is permitted to access, download, copy, distribute, and use these materials in any way, even commercially, with proper attribution. For these reasons, we cannot publish previously copyrighted maps or satellite images created using proprietary data, such as Google software (Google Maps, Street View, and Earth). For more information, see our copyright guidelines: http://journals.plos.org/plosone/s/licenses-and-copyright.

You may seek permission from the original copyright holder of Figure 1 to publish the content specifically under the CC BY 4.0 license. 

If you are unable to obtain permission from the original copyright holder to publish these figures under the CC BY 4.0 license or if the copyright holder’s requirements are incompatible with the CC BY 4.0 license, please either i) remove the figure or ii) supply a replacement figure that complies with the CC BY 4.0 license. Please check copyright information on all replacement figures and update the figure caption with source information. If applicable, please specify in the figure caption text when a figure is similar but not identical to the original image and is therefore for illustrative purposes only.

Reviewers' comments:

Reviewer's Responses to Questions

**Comments to the Author**

1. Is the manuscript technically sound, and do the data support the conclusions?

Reviewer #1: Yes

Reviewer #2: Yes

Reviewer #3: Yes

Reviewer #4: Partly

2. Has the statistical analysis been performed appropriately and rigorously? 

Reviewer #1: Yes

Reviewer #2: Yes

Reviewer #3: Yes

Reviewer #4: I Don't Know

3. Have the authors made all data underlying the findings in their manuscript fully available?

Reviewer #1: Yes

Reviewer #2: Yes

Reviewer #3: Yes

Reviewer #4: Yes

4. Is the manuscript presented in an intelligible fashion and written in standard English?

Reviewer #1: Yes

Reviewer #2: Yes

Reviewer #3: Yes

Reviewer #4: Yes

5. Review Comments to the Author

Reviewer #1: Thank you for giving me the opportunity to review this paper about thyroid carcinoma in children, adolescents, and young adults in Brazil. The authors are to be congratulated for performing a clinically relevant study concerning real world data of 11 cancer registries in Brazil.

Introduction

The authors should put the informationof risk factors and genetic alterations (line 220-266) in the introduction section, as the introduction should provide information about thyroid cancer in adolescents and young adults to the reader. Otherwise, the paper is hard to follow and lacks of a logical flow. This would also help to shorten the dicussion section.

Materials and methods

Statement of ethical approval is not applicable

Joinpoint regression is a feasible and commonly used method for analyzing cancer trends.

Discussion

The authors should use the discussion section to clearly present their results concerning the different regions of Brazil. In its current form, the discussion section is too lengthy and the authors may divide the discussion into section headings.

There should be a section about gender differences.

Reviewer #2: Authors should discuss the following in the revised manuscript:

To see whether the increase in incidence was due to overdiagnosis through screening, authors should have stratified the sample by stage at diagnosis.

If they found higher early stage at diagnosis, then the likelihood of the increase in incidence rate is due to screening becomes more likely.

If authors have no access to stage at diagnosis, they should refer readers to other studies that have investigated trends in stage at diagnosis during same time period. Then they could speculate according to comparisons.

Also, authors should speculate that the increase could be due to exposure to radiation at dental office (they can add references). They also should refer to other studies to at least shed lights on possible etiology. For instance, is the southern region more developed areas of the countries where access to dental services is higher than other region and that’s why residents are more likely to get exposed to risk factors? The same thing apply to Iodine exposure.

I’d also suggest authors to include graphs about linear changers (AAPC) over time.

By looking at the location of the selected registries, most of them from are from southeast region which contains a very large population compare to the rest of the regions. It will be good discuss this point in the discussion too.

Finally, authors should add limitation section and enlist all that apply.

Reviewer #3: PONE-D-19-34167: Thyroid carcinoma in children, adolescents, and young adults in Brazil: A report from 11 population-based cancer registries

Summary: The study by de Camargo et al. examined the incidence and trends of thyroid cancer among children (0-14 years) and young adults (15-39 years) in Brazil. The investigators analyzed data from 11 population-based cancer registries. They calculated age-adjusted incidence rates (AAIR), age specific incidence rates (ASIR), and changes in incidence trends between 2000-2013 using jointpoint regression analyses. The results show an AAIR of 42 cases per a million persons. The incidence rate was higher in females than males. Participants of age 15-39 years had higher incidence rate than those of age 0-14. The trend analysis also increasing incidence of thyroid cancer in Brazil between 2000 and 2013. Overall, this is an important study that addresses a knowledge gap of the patterns of thyroid cancer incidence in the Brazilian population. Below are suggestions for improvement.

1. Why did the investigators express the incidence rate per one million persons? Typically, incidence rates are expressed per 100,000 population. The use of a million-population estimate will make it difficult to compare the incidence rate with that of other populations that have used the typical 100,000 population. The investigators should consider revising the estimated population rate to 100,000 instead of a million.

2. Data from 11 population-based cancer registries were analyzed. Why were these registries chosen? What proportion of the Brazilian population do these registries represent? The concern here is generalizability of the findings to the broader Brazilian population.

3. For the subgroup comparisons, e.g., comparison between males and females, please provide p-values showing significant differences between groups. In other words, is the estimate for females significantly different from that of males? We know that the estimated values are different but are the estimates significantly different from each other?

4. “AYAs had a 40-fold greater incidence than children (0–14 years).” What values were used to calculate the 40-fold difference?

5. In the abstract, please indicate the age ranges used to classify participants into children vs. older adults for easy comprehension.

6. Abstract, please correct “11.081” as 11,081.

7. The ICD-02 and 03 codes used to identify the thyroid cancer cases, are these the standard codes? Can you cite a reference or two indicating the appropriateness of these codes?

8. Death certificates are not reliable for classifying cancer cases. Please remove the 4 cases determined by death certificates alone from the analyses.

9. Some of the Tables don’t fit into the manuscript space (e.g., Table 4). Please reformat the table, possibly using smaller font size.

10. A link to a pdf document was provided in the Discussion but the document appears to be in Portuguese. Is there an English version of the document?

Reviewer #4: This kind of manuscript, which carries demographic data of Brazilian population without any molecular identification, therefor this report will useful for national journal.

This kind of study need to more case and case control for analysis as age, sex and disease match.

6. PLOS authors have the option to publish the peer review history of their article (what does this mean?). If published, this will include your full peer review and any attached files.

Reviewer #1: No

Reviewer #2: No

Reviewer #3: Yes: Samuel O. Antwi

Reviewer #4: No

---

## [Author Response · Author response to Decision Letter 0]

25 Mar 2020

Reviewer #1: Thank you for giving me the opportunity to review this paper about thyroid carcinoma in children, adolescents, and young adults in Brazil. The authors are to be congratulated for performing a clinically relevant study concerning real world data of 11 cancer registries in Brazil.

Introduction

The authors should put the information of risk factors and genetic alterations (line 220-266) in the introduction section, as the introduction should provide information about thyroid cancer in adolescents and young adults to the reader. Otherwise, the paper is hard to follow and lacks of a logical flow. This would also help to shorten the dicussion section.

R: Thank you for the comment, we add risk factors in the Introduction and shorten the discussion.

Added to introduction (lines 75-85): “The prevalence of incidental thyroid abnormalities detected by ultrasound in Korean adults is about 13–67% [12]. In children, few reports have studied the prevalence of ultrasound-detected findings in the thyroid, with the exception of screening programs conducted around Chernobyl [13]. A survey including thyroid ultrasound examinations for children aged ≤18 years was conducted in Japanese children unaffected by radioactive material from the Fukushima Nuclear Power Plant accident, and thyroid nodules were identified in 1.6% of participants; 99% were classified as ‘no further examination required’ [14, 15]. Neck ultrasound done for other reasons detected incidental thyroid abnormalities in 18% of children, and the majority were cystic lesions. No malignancy was found in this group of patients [16].”

We deleted from discussion:

(line 241): “High iodine intake, obesity, and high birth weight are risk factors for thyroid cancer.”

(lines 259-265): “In a recent evaluation of urinary iodine concentrations in schoolchildren from all Brazilian regions, the highest concentration was seen in the Northeast region (298.80µg/L). In the Amazon state located in the North region, the median urinary iodine concentration was low (197.60 µg/L). One-fifth of the children had adequate intake (20.4%) presenting urinary iodine concentrations between 100 and199 µg/L, while 25.2% had more than adequate intake and 44.6% had excessive iodine intake (≥ 300 µg/L)”

(lines 283-285): “In several countries, the prevalence of obesity has almost doubled: in Israel, it increased from 5.8% in 1975 to 11.9% in 2016, Andorra from 6.2% to 12.8%, and Malta from 7.4% to 13.4% [32].”

Materials and methods

Statement of ethical approval is not applicable

R: Included in Material and Methods: “All data used for this analysis were public access data and thus that ethical approval this study was not required.”

Discussion

The authors should use the discussion section to clearly present their results concerning the different regions of Brazil. In its current form, the discussion section is too lengthy and the authors may divide the discussion into section headings.

There should be a section about gender differences.

R: We shorten the discussion as describe above. It was included gender differences. 

Reviewer #2: Authors should discuss the following in the revised manuscript:

To see whether the increase in incidence was due to overdiagnosis through screening, authors should have stratified the sample by stage at diagnosis.

If they found higher early stage at diagnosis, then the likelihood of the increase in incidence rate is due to screening becomes more likely.

If authors have no access to stage at diagnosis, they should refer readers to other studies that have investigated trends in stage at diagnosis during same time period. Then they could speculate according to comparisons.

R: We agree with the reviewer and recognize the relevance of the information about the stage and size tumor, but unfortunately, we do not have data, and it was included as a major limitation.

Also, authors should speculate that the increase could be due to exposure to radiation at dental office (they can add references). They also should refer to other studies to at least shed lights on possible etiology. For instance, is the southern region more developed areas of the countries where access to dental services is higher than other region and that’s why residents are more likely to get exposed to risk factors? The same thing apply to Iodine exposure.

R: Unfortunately, we do not have precise data of risk factors and the best we could provide we included in the discussion: “Available data has shown increased of dental x-ray of 1% during the period 2008 to 2019 among children, adolescents and young adults (http://www2.datasus.gov.br/).” 

I’d also suggest authors to include graphs about linear changers (AAPC) over time.

R. Figure three includes AAPC over time (lines 194-195).

By looking at the location of the selected registries, most of them from are from southeast region which contains a very large population compare to the rest of the regions. It will be good discuss this point in the discussion too.

R: We agree with the reviewer and included in the Discussion: “A major limitation of this study is that there is no information regards size and stage of tumors. Another major limitation is that risk factors as use of radiation, is unknown. These 11 PBCR cover approximately 20% of the total Brazilian population so we cannot generalize this data to entire Brazil.”

Finally, authors should add limitation section and enlist all that apply.

R: See above. 

Reviewer #3: PONE-D-19-34167: Thyroid carcinoma in children, adolescents, and young adults in Brazil: A report from 11 population-based cancer registries

Summary: The study by de Camargo et al. examined the incidence and trends of thyroid cancer among children (0-14 years) and young adults (15-39 years) in Brazil. The investigators analyzed data from 11 population-based cancer registries. They calculated age-adjusted incidence rates (AAIR), age specific incidence rates (ASIR), and changes in incidence trends between 2000-2013 using jointpoint regression analyses. The results show an AAIR of 42 cases per a million persons. The incidence rate was higher in females than males. Participants of age 15-39 years had higher incidence rate than those of age 0-14. The trend analysis also increasing incidence of thyroid cancer in Brazil between 2000 and 2013. Overall, this is an important study that addresses a knowledge gap of the patterns of thyroid cancer incidence in the Brazilian population. Below are suggestions for improvement.

1. Why did the investigators express the incidence rate per one million persons? Typically, incidence rates are expressed per 100,000 population. The use of a million-population estimate will make it difficult to compare the incidence rate with that of other populations that have used the typical 100,000 population. The investigators should consider revising the estimated population rate to 100,000 instead of a million.

R: In children, adolescents and young adults the incidence rate are more usually express per million persons in all paper because of rarity. 

2. Data from 11 population-based cancer registries were analyzed. Why were these registries chosen? What proportion of the Brazilian population do these registries represent? The concern here is generalizability of the findings to the broader Brazilian population.

R: Among the eleven PBCR eight cancer registries data were included in the Vol. X or XI of CI5C, the main publication of the WHO on cancer incidence, the other registries follow the same rules of the IARC for an exhaustive and of good quality registration [Câncer no Brasil: dados dos registros de base populacional, vol. 4, INCA 2010. Available from: https://www.inca.gov.br/publicacoes/livros/cancer-no-brasil-dados-dos-registros-de-base-populacional; Bray F, Colombet M, Mery L, Piñeros M, Znaor A, Zanetti R and Ferlay J, editors (2017) Cancer Incidence in Five Continents, Vol. XI (electronic version). Lyon: International Agency for Research on Cancer. Available from: http://ci5.iarc.fr]. 

These 11 PBCRs cover approximately 20% of the total Brazilian population.

These was included in the limitations.

3. For the subgroup comparisons, e.g., comparison between males and females, we included female:male ratio in table 4.

4. “AYAs had a 40-fold greater incidence than children (0–14 years).” What values were used to calculate the 40-fold difference?

R: The values were in table 4 – Crude incidence rate among children 0-14 (1.87) and curde incidence rate among AYA (80.28)

5. In the abstract, please indicate the age ranges used to classify participants into children vs. older adults for easy comprehension.

R: It was included.

6. Abstract, please correct “11.081” as 11,081.

R: It was corrected.

7. The ICD-02 and 03 codes used to identify the thyroid cancer cases, are these the standard codes? Can you cite a reference or two indicating the appropriateness of these codes?

R: Yes the ICD-02 and ICD-03 are the International code of oncology disease. We included the reference.

8. Death certificates are not reliable for classifying cancer cases. Please remove the 4 cases determined by death certificates alone from the analyses.

R: About the DCO, rules from International Agency for Research on Cancer (IARC) recommends to included them.

9. Some of the Tables don’t fit into the manuscript space (e.g., Table 4). Please reformat the table, possibly using smaller font size.

R: Thanks for notice we corrected all tables.

10. A link to a pdf document was provided in the Discussion but the document appears to be in Portuguese. Is there an English version of the document?]

R: No, unfortunately the document is only in Portuguese.

Reviewer #4: This kind of manuscript, which carries demographic data of Brazilian population without any molecular identification, therefor this report will useful for national journal.

This kind of study need to more case and case control for analysis as age, sex and disease match.

R: Unfortunately there are few papers that includes molecular identification on incidence rates.

---

## [Decision Letter · Decision Letter 1]

15 Apr 2020

Thyroid carcinoma in children, adolescents, and young adults in Brazil: A report from 11 population-based cancer registries

PONE-D-19-34167R1

Dear Dr. de Camargo,

We are pleased to inform you that your manuscript has been judged scientifically suitable for publication and will be formally accepted for publication once it complies with all outstanding technical requirements.

With kind regards,

Amir Radfar, MD,MPH,MSc,DHSc

Academic Editor

PLOS ONE

Additional Editor Comments (optional):

Reviewers' comments:

Reviewer's Responses to Questions

**Comments to the Author**

1. If the authors have adequately addressed your comments raised in a previous round of review and you feel that this manuscript is now acceptable for publication, you may indicate that here to bypass the “Comments to the Author” section, enter your conflict of interest statement in the “Confidential to Editor” section, and submit your "Accept" recommendation.

Reviewer #1: All comments have been addressed

Reviewer #2: All comments have been addressed

2. Is the manuscript technically sound, and do the data support the conclusions?

Reviewer #1: Yes

Reviewer #2: Yes

3. Has the statistical analysis been performed appropriately and rigorously? 

Reviewer #1: Yes

Reviewer #2: Yes

4. Have the authors made all data underlying the findings in their manuscript fully available?

Reviewer #1: Yes

Reviewer #2: Yes

5. Is the manuscript presented in an intelligible fashion and written in standard English?

Reviewer #1: Yes

Reviewer #2: Yes

6. Review Comments to the Author

Reviewer #1: Please check the manuscript for spelling mistakes, eg table 2 "others carcinomas" instead of "other".

Reviewer #2: I would like to thank the authors for chosing PLOS ONE for publishing their results. I think they gave resonable justifications for all asked questions.

7. PLOS authors have the option to publish the peer review history of their article (what does this mean?). If published, this will include your full peer review and any attached files.

Reviewer #1: No

Reviewer #2: No

---

## [Editor Report · Acceptance letter]

21 Apr 2020

PONE-D-19-34167R1 

Thyroid carcinoma in children, adolescents, and young adults in Brazil: A report from 11 population-based cancer registries 

Dear Dr. de Camargo:

I am pleased to inform you that your manuscript has been deemed suitable for publication in PLOS ONE. Congratulations! Your manuscript is now with our production department. 

With kind regards,

on behalf of

Dr. Amir Radfar

Academic Editor

PLOS ONE